# Features of mobile apps for diabetic kidney disease self-management: A scoping review

Hasmawati Yahya[1,2�], Nani Draman[2*‡], Azidah Abdul Kadir[2�], Najib Majdi Yaacob[3‡]

**1** Nursing Department, Kolej Poly-Tech MARA, Kota Bharu, Kelantan, Malaysia, **2** Department of Family Medicine, School of Medical Science, Health Campus, Universiti Sains Malaysia Kubang Kerian, Kelantan, Malaysia, **3** Department of Biostatistics and Research Methodology, Health Campus, Universiti Sains Malaysia, Kubang Kerian, Kelantan, Malaysia

MOON These authors contributed equally to this work
‡ These authors also contributed equally to this work.
* drnani@usm.my

## Abstract

### Background

Diabetic kidney disease (DKD) is a chronic complication of diabetes mellitus (DM). DKD and chronic kidney disease (CKD) are both long-term conditions requiring ongoing patient care. Individuals living with DKD or CKD can benefit from mobile apps that support effective self-management. However, limited evidence is available about what mobile apps features are commonly included for DKD.

### Objective

This scoping review aimed to identify the features of mobile applications on self-management for individuals with DKD, DM or CKD.

### Methods

The review followed the Joanna Briggs Institute (JBI) methodology for scoping reviews and adhered to PRISMA-ScR reporting guidelines. Five databases (PubMed, Scopus, SAGE Journals, ScienceDirect, and Web of Science) were searched from inception to February 26, 2025. Studies were included if they reported on mobile apps supporting self-management in adults with DKD, DM or CKD. DM and CKD apps were included due to similar self-management needs such as blood sugar or blood pressure tracking that are also relevant to people with DKD. Data were extracted on study characteristics, app features, use of technology, integration with care teams, and reported outcomes.

### Results

Out of 3521 records identified, eleven studies met the inclusion criteria. Five studies focused on CKD, two on DKD, and four on diabetes. Across the eleven mobile apps

**Data availability statement:** All relevant data are within the paper and its Supporting Information files.

**Funding:** This work was supported by the Universiti Sains Malaysia, Research University Team (RUTeam) Grant Scheme with Project No: 1001/PPSP/8580083, Project Code: TE0034 (Reference No: 2022/0501] The funding provided covered the author's role as a research assistant and facilitated access to the Universiti Sains Malaysia library's electronic databases and resources.

**Competing interests:** The authors have declared that no competing interests exist.

reviewed, four core domains of self-management were identified: self-care monitoring (91%), educational components (64%), patient support and motivation (100%), and performance incentives (18%). Four apps employed wearable devices and incorporated supportive devices such as Bluetooth glucometers. However, only two apps included real-time communication features with providers integration with healthcare teams and gamification strategies.

## Conclusions

Mobile apps targeting DKD frequently incorporate monitoring, education, and motivational features. However, consistent integration with healthcare providers and incentive-based engagement strategies remains limited. Future app development should emphasise personalised feedback, clinical integration, and sustained engagement mechanisms to enhance usability and impact.

---

### Introduction

Diabetes and chronic kidney disease (CKD) are two significant global health challenges that often coexist and interact. Diabetic kidney disease (DKD), a condition that emerges when diabetes leads to progressive kidney damage, is one of the most serious long-term complications of diabetes. It significantly increases the risk of kidney failure, cardiovascular disease, and premature death. Managing these conditions requires patients to take on various daily tasks, such as monitoring blood glucose or blood pressure, adhering to medications, making dietary changes, and engaging in healthy lifestyle behaviours. These self-management responsibilities can be overwhelming for many individuals, especially without structured support [1,2].

In recent years, mobile applications (apps) have gained attention as practical and scalable tools for self-management of chronic diseases like diabetes and CKD. These apps offer a variety of features that can help patients better understand their condition, track health metrics, receive medication or appointment reminders, and access educational content on diet, exercise, and symptoms. Because mobile phones are widely accessible and integrated into many people's daily lives, mobile apps have become a convenient and potentially impactful way to enhance disease self-care [3].

Self-management refers to an individual's ability to manage the symptoms, treatment, lifestyle changes, and psychosocial consequences of living with a chronic condition. It emphasizes active participation in care, including making informed decisions, adhering to treatment plans, and engaging in behaviors that promote health and well-being [4,5]. In the context of diabetic kidney disease and chronic kidney disease, effective self-management is linked to improved clinical outcomes, greater self-efficacy, and reduced healthcare utilization [6,7].

Despite the growing number of mobile apps developed for diabetes or CKD individually, relatively few are designed specifically for people with DKD. Because DKD shares key self-management needs with both diabetes and CKD, mobile apps developed for either condition often include features relevant to DKD management,

such as blood glucose and blood pressure monitoring, medication reminders, educational content, and provide communication tools [3,8]. Including mobile apps targeting diabetes, CKD, and DKD in this review enables a more comprehensive assessment of how digital health tools support self-management across these interrelated conditions [11–21].

Therefore, this scoping review aims to explore and map the key features of mobile apps developed to support self-management for people living with diabetes, CKD, or DKD. Examining mobile applications across these three interrelated conditions provides a more comprehensive understanding of how mobile technology supports key self-management behaviours, including monitoring, education, motivation, and treatment adherence. By understanding what's already available and how these features are used, this review aims to highlight areas where mobile app development is well established and where more tailored, patient-centred solutions may still be needed [2,3].

## Materials and methods

This scoping review was conducted in accordance with the Joanna Briggs Institute (JBI) methodology for scoping reviews and reported following the Preferred Reporting Items for Systematic Reviews and Meta-Analyses extension for Scoping Reviews (PRISMA-ScR) checklist. The research questions in this scoping review were developed using the Joanna Briggs Institute's Population–Concept–Context (PCC) framework to ensure alignment with scoping review methodology [S2 Table]. The population of interest includes individuals diagnosed with both diabetes and chronic kidney disease (CKD) or diabetic kidney disease (DKD). The concept explored is self-management, which refers to patients' active participation in managing their health conditions. The context focuses on the use of mobile apps that support these self-management activities. The PCC elements guiding this review are summarized in Table 1.

### Protocol and registration

No protocol was registered for this scoping review. However, the methodology followed the Joanna Briggs Institute (JBI) guidelines for conducting scoping reviews and adhered to the PRISMA-ScR (Preferred Reporting Items for Systematic Reviews and Meta-Analyses extension for Scoping Reviews) reporting checklist. The JBI framework was selected over other scoping review approaches (such as the Arksey and O'Malley framework) because it provides comprehensive and up-to-date methodological guidance, including the use of the Population–Concept–Context (PCC) framework and structured data charting procedures.

### Inclusion criteria

Studies were included if they described or evaluated mobile applications (apps) designed to support self-management among individuals with diabetic kidney disease (DKD) or diabetes or chronic kidney disease. Articles must report on at least one self-management-related feature integrated into a mobile app, such as health monitoring, patient education, motivational tools, or performance feedback. Only studies published in English were included. There were no publication year, study design, or geographic location restrictions.

More specifically, the inclusion criteria were:

**Table 1. PCC Framework for the scoping review research questions.**

| PCC element | Keywords/Terms |
| --- | --- |
| Population | Diabetes AND (Chronic Kidney Disease OR Diabetic Kidney Disease) |
| Concept | Self-management |
| Context | Mobile applications (mobile apps) |

i. Population: Adults diagnosed with type 2 diabetes and chronic kidney disease (CKD) or diabetic kidney disease (DKD). Studies focusing on patients with both conditions, or where DKD is clearly defined, will be included.

ii. Concept: The main focus is self-management interventions delivered via mobile applications. This includes features such as self-care monitoring, education, motivation, support, and performance incentives.

iii. Context: Studies conducted in any healthcare or community setting, with mobile applications (not limited by platform) used to support self-management among DKD patients. No geographic restrictions exist, as long as the study includes relevant app-based interventions.

iv. Types of Studies: This scoping review included interventional studies (both randomised and non-randomised), observational studies (such as cohort and cross-sectional designs), and studies describing the development phase of mobile apps. Protocol papers and conference abstracts were excluded. Studies focusing on healthcare personnel, caregivers, or family members rather than patients were also excluded.

## Information sources

Five electronic databases were systematically searched: PubMed, Scopus, SAGE Journals, ScienceDirect, and Web of Science. In addition, reference lists of the included studies were also manually screened to identify additional eligible studies; however, none met the inclusion criteria.

## Search strategy

A comprehensive search strategy has been developed using keywords and Medical Subject Headings (MeSH) related to diabetes, chronic kidney disease, diabetic kidney disease, self-management, and mobile applications. Each database has a tailored search strategy, and Boolean operators (AND/OR) were used to refine the search queries. All databases were searched from inception to 26th February 2025, with no limits on language set. An example of the whole search string is included in S3 Table.

## Study selection and data charting

All search results were imported into reference management software, Zotero, and duplicates were removed. Two reviewers independently screened titles and abstracts against the inclusion criteria. Full-text articles of potentially relevant studies were reviewed independently. Data charting was conducted using a standardised extraction form developed by the review team, which included predefined fields based on the review objectives and PCC framework. Discrepancies between reviewers have been resolved through discussion or consultation with a third reviewer when necessary.

## Data extraction and synthesis

Relevant data were extracted into a structured form, including study characteristics, target population, mobile app features, technologies used, outcomes measures, and integration with care teams, if available. The data extraction form was developed by the review team based on the review objectives and the PCC framework, and was tested and refined during the initial stages of data extraction to ensure clarity, consistency, and reproducibility, in line with JBI guidance for scoping reviews.

Features identified from the included mobile apps were categorised into four functional domains: self-care monitoring, education, motivation and patient support, and performance incentives. Categorisation was guided by the primary function of each feature as described in the included studies and mapped to the predefined domain definitions developed by the review team. Where a feature addressed more than one function, it was assigned to the domain that best reflected its main intended purpose. Medication-related functionalities (such as medication reminders, adherence tracking, or insulin

 

monitoring) were categorized under the self-care monitoring domain. Any uncertainties in categorisation were discussed among the review team and resolved by consensus.

## Data items

The following data items have been charted:

i. Author(s), year of publication, and country of study

ii. Study design and setting

iii. Target population (e.g., type 2 diabetes with chronic kidney disease)

iv. Mobile apps name

v. Core functional and details of self-management features

vi. Technology used and usage of supportive technologies (e.g., sensors or wearables)

vii. Integration with care team and

viii. Outcomes measurement used

## Quality assessment

Consistent with the scoping review methods and objective of this scoping review which are to identify and map the key features of mobile applications used for self-management among individuals with diabetic kidney disease, this study focused on providing a broad overview of the available evidence [9,10]. As per JBI guidelines, critical appraisal was not conducted, in line with the exploratory objectives of the review.

# Results

A total of 11 studies [11–21] met our review criteria. Fig 1 provides a flow diagram of article identification and inclusion.

Table 2 summarises the main characteristics of the 11 studies included in this scoping review. These studies varied in terms of country of origin, year of publication, study design, target population, and the specific self-management features integrated into the mobile applications. Several studies also reported the use of supportive devices to enhance the app's functionality. This overview provides context on the diversity of approaches employed in designing and implementing mobile health interventions for patients with diabetes, CKD, or DKD across different healthcare settings.

## Study methods

The studies included in this review employed various research designs. Five studies utilized a randomized clinical trial (RCT) design [12–14,17,21] while three studies adopted user-centered or co-development approaches [11,15,20]. One study conducted a pilot validation [16] another employed a prospective cohort design [19] and one study carried out a descriptive technical validation [18].

## Year of publication and target population

The included studies were published between 2014 and 2024, with a concentration of research conducted in the past five years from 2020 to 2024. The earliest study was by Hidalgo et al. [18] in 2014, while the most recent were published in 2024. The studies targeted patients with a range of chronic conditions, primarily focusing on chronic kidney disease (CKD), diabetic kidney disease (DKD), and type 2 diabetes mellitus (T2DM). As shown in Fig 2 specifically, five studies involved CKD patients [11,12,15,19], two focused on DKD patients [14,16] and four studies targeted individuals with T2DM

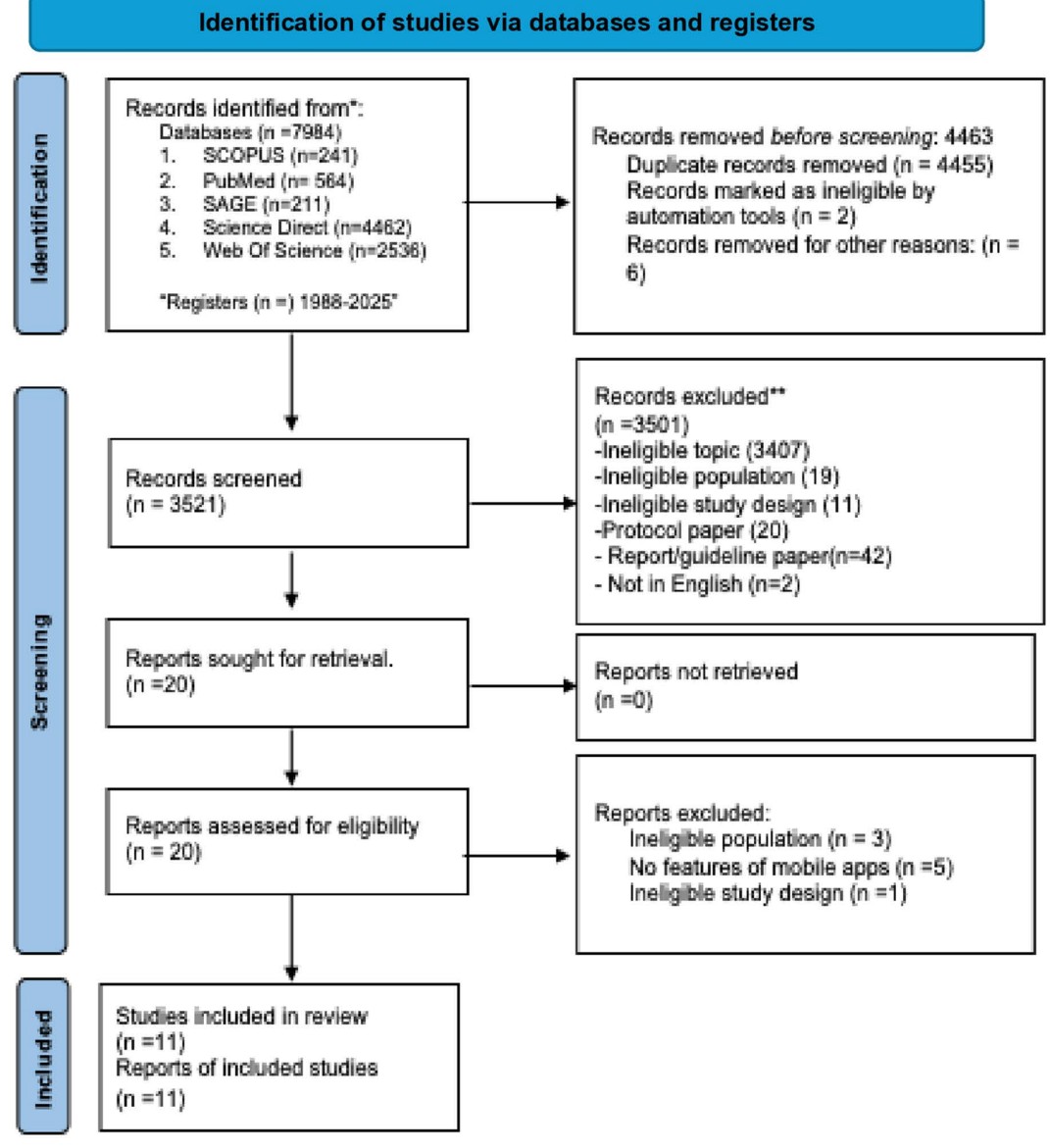

**Fig 1. Flow diagram of article selection.**

[17,18,20,21]. One study [18] included both T1DM and T2DM patients. This diversity in target populations reflects the overlapping and interrelated nature of these chronic conditions in mobile health research.

### Core functional domains of self-management features

Table 3 summarizes four recurring core functional domains of self-management features identified across the included studies. As shown in Table 3, self-care monitoring was present in ten studies, allowing users to track parameters such as blood glucose, blood pressure, and physical activity. Educational components were featured in seven studies, providing patients with disease-specific knowledge and lifestyle guidance. All eleven studies included patient support and

**Table 2. Characteristics of included studies.**

| Study No. | Article | Country | Year | Study design | Target population | Main self-management features | Other supportive devices used |
|---|---|---|---|---|---|---|---|
| 1. | Lightfoot et al. [11] | Canada | 2022 | Intervention co-development | CKD patients | i. Self-care monitoring<br>ii. Educational<br>iii. Patient's support/ motivational | None reported |
| 2. | Ong et al. [12] | United States | 2021 | A randomised clinical trial | CKD patients | i. Self-care monitoring<br>ii. Patient's support/ motivational | None reported |
| 3. | Li et al. [13] | Taiwan | 2020 | A randomised clinical trial | CKD patients | i. Self-care monitoring<br>ii. Patient's support/ motivational | i. Heart rate wristband (detect steps, calories, sleep)<br>ii. LINE app group – to deliver diet and exercise information and teleconsultations |
| 4. | Waki et al. [14] | Japan | 2024 | A randomised clinical trial | DKD Patients | i. Self-care monitoring<br>ii. Patient's support/ motivational | Bluetooth glucometer |
| 5. | Markossian et al. [15] | United States | 2021 | User-centred design-development study | CKD patients | i. Self-care monitoring<br>ii. Educational<br>iii. Patient's support/ motivational<br>iv. Performance incentive | None reported |
| 6. | Toapanta et al. [16] | Spain | 2024 | Validation-Pilot monitoring study | DKD Patients | i. Self-care monitoring<br>ii. Educational<br>iii. Patient's support/ motivational | chat message/video call |
| 7. | Zhang et al. [17] | China | 2024 | A randomised clinical trial | T2DM | i. Self-care monitoring<br>ii. Educational<br>iii. Patient's support/ motivational<br>iv. Performance incentive | None reported |
| 8. | Hidalgo et al. [18] | Spain | 2014 | System Design & Technical Validation (Descriptive study) | T1DM T2DM | i. Self-care monitoring<br>ii. Educational<br>iii. Patient's support/ motivational | None reported |
| 9. | Tsai et al. [19] | Switzerland | 2021 | Prospective Cohort Study | CKD patients | i. Self-care monitoring<br>ii. Educational<br>iii. Patient's support/ motivational | None reported |
| 10. | Berlot et al. [20] | New York, US. | 2024 | Development and evaluation study - single-arm trial study | T2DM | i. Educational<br>ii. Patient's support/ motivational | None reported |
| 11. | Gunawardena et al. [21] | Sri Lanka | 2019 | A randomized clinical trial | T2DM | i. Self-care monitoring<br>ii. Patient's support/ motivational | Bluetooth glucometer |

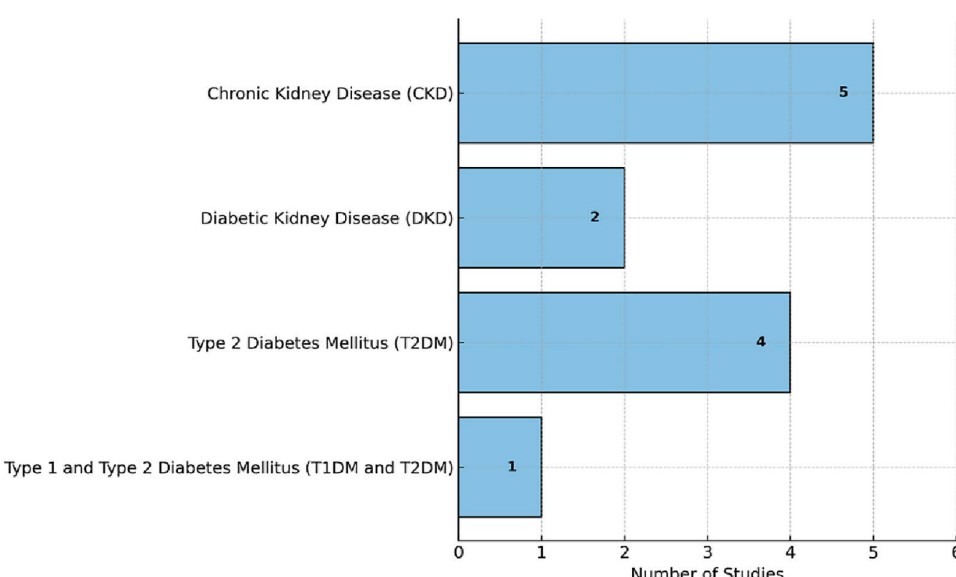

**Fig 2. Number of studies by condition.**

**Table 3. Core functional domains across the 11 studies.**

| Study | Self-care Monitoring | Educational | Patient Support/ Motivation | Performance Incentive |
|---|---|---|---|---|
| Study 1 [11] | Yes | Yes | Yes | No |
| Study 2 [12] | Yes | No | Yes | No |
| Study 3 [13] | Yes | No | Yes | No |
| Study 4 [14] | Yes | No | Yes | No |
| Study 5 [15] | Yes | Yes | Yes | Yes |
| Study 6 [16] | Yes | Yes | Yes | No |
| Study 7 [17] | Yes | Yes | Yes | Yes |
| Study 8 [18] | Yes | Yes | Yes | No |
| Study 9 [19] | Yes | Yes | Yes | No |
| Study 10 [20] | No | Yes | Yes | No |
| Study 11 [21] | Yes | No | Yes | No |
| **Frequency (%)** | **10 (90.9%)** | **7(63.6%)** | **11(100%)** | **2(18.1%)** |

motivational features, such as reminders, feedback, and communication tools. Lastly, performance incentive features, such as goal tracking and gamification, were incorporated in two studies [15,17] aimed at encouraging adherence and engagement.

Further details of the four key self-management domains identified across the included studies are presented in Table 4. These domains reflect the core functionalities integrated into mobile health interventions aimed at enhancing chronic disease self-management. Among the 11 studies, patient support and motivational features were universally implemented. These included reminders, personalized feedback, peer interactions, and clinician alerts tools known to enhance adherence and user engagement. Self-care monitoring was nearly as prevalent, appearing in 10 studies [11–21] with features such as self-monitoring blood glucose, blood pressure, pulse and activity tracking, which some supported by wearable technology.

**Table 4. Details of core functional domains for self-management, integration, and outcome measures.**

| No. | Study | Mobile apps name | Core functional domains of self-management features description | | | Performance incentive | Integration with care team | | Outcome measures |
|---|---|---|---|---|---|---|---|---|---|
| | | | Self-care | Educational | Patient support and motivation | | Availability | Details | |
| 1 | Lightfoot et al. [11] | My Kidney & Me (MK&M) | i.Blood pressure (BP)<br>ii.Step counting<br>iii.Healthy eating<br>iv.Goal setting features | -Videos:<br>i. The kidneys<br>ii. Kidney disease<br>iii. Kidney disease and general health<br>iv.Treatment options available<br>v. Reducing my health risks<br>vi. Moving more and being active<br>vii. Keeping my muscles healthy<br>viii. Eating a healthy balanced diet<br>ix. Managing my symptoms<br>x. Improving my sleep quality<br>xi. Looking after my well being<br>xii. Goal setting<br>-Learning session<br>-Quizzes | Tailored education, goal setting, behavior change support, patient forum | None reported | Yes | Minimal—no real-time communication or data sharing with healthcare providers. | Adherence, blood pressure(BP) control |
| 2 | Ong et al. [12] | e-KidneyCare | i. Blood pressure<br>ii.Assessing symptoms<br>iii. Connected with pharmacy database<br>iv. Blood test: eGFR, hemoglobin, potassium, phosphate | None | Personalized feedback, lab and symptom tracking, provider alerts | None reported | Yes | Integrated alert system data from the app triggered alerts to healthcare teams when thresholds were exceeded. | HbA1c reduction, Adherence |
| 3 | Li et al. [13] | WowGoHealth app | i. Blood pressure<br>ii.Heart rate<br>iii.Steps counting | Through support group- none in app | Peer support via social media group, personalized advice, goal tracking | None reported | Yes | Moderate integration: data analytics provided trend summaries for care teams feedback. | Weight loss, BP, HbA1c |
| 4 | Waki et al. [14] | DialBetesPlus | i.Step counts<br>ii.Blood glucose<br>iii. Blood pressure<br>iv.Body weight<br>v.Dietary evaluation feedback | None | Self-monitoring with feedback, lifestyle guidance, in-app feedback | None reported | Yes | Data synchronized with electronic medical records (EMR) and viewable by physicians during appointments. | Glucose level, adherence |

*(Continued)*

| No. | Study | Mobile apps name | Core functional domains of self-management features description | | | Performance incentive | Integration with care team | | Outcome measures |
|---|---|---|---|---|---|---|---|---|---|
| | | | Self-care | Educational | Patient support and motivation | | Availability | Details | |
| 5 | Markossian et al. [15] | N/A | i. Blood pressure ii.Weight iii.Blood glucose iv.Medication reminder | Information on i. CKD ii. Nutrition and recommended diets iii. Concept of self-management iv. Anxiety and depression v. Covid-19 symptoms and resources vi. Symptoms of CKD | Health tracking with feedback, real-time provider interaction, education | Gamified or performance-based incentives (non-financial) | No | No integration: app was patient-facing only, focused on goal setting and self-motivation. | Goal adherence, weight loss |
| 6 | Toapanta et al. [16] | NORA application | i.Height/weight ii. Self monitoring blood glucose (SMBG) iii.SpO2 iv. Blood pressure v. Steps count vi. Medication reminder | Health education material: i. Education module- managing blood pressure, blood glucose and weight. Medication adherence ii. FAQ | Mood check-ins, clinician messaging, educational info, adherence support | None reported | Planned for future | No integration—developmental phase, care team involvement not yet implemented. | App usability |
| 7. | Zhang et al. [17] | SMARTDiabetes | i. Blood pressure ii.Body mass index iii. Medication reminder iv. Blood test: Glucose, LDL, Cholesterol, HbA1c | The information is organized into seven sections: I. General noncommunicable diseases, ii.Diabetes, iii. Hypertension, iv.Weight control, v.Physical exercise, vi.Other lifestyle vii.Physical examination. | Personalized goals, reminders, family support, education | Small incentives for participation (gift cards, social recognition) | Yes | Real-time data synced to central dashboard accessible by care team to monitor patient progress. | BP, HbA1c, app adherence |
| 8. | Hidalgo et al. [18] | gIUCModel | i.Diet ii.Sport activity (scale level of exercise 1–10+introduce the duration of the exercise and complete a remark field explaining the sport activity performed.) iii.Weight iv..Body mass index v. Insulin reminder vi. Blood test- glucose level | Information on i.Hypoglycaemia management ii. Diet iii. Symptoms | Personalized recommendations, education, feedback engine | None reported | Planned for future | Planned for future—currently no integration but designed with provider-facing potential. | Usability, acceptability |

*(Continued)*

| No. | Study | Mobile apps name | Core functional domains of self-management features description | | | Performance incentive | Integration with care team | | Outcome measures |
| --- | --- | --- | --- | --- | --- | --- | --- | --- | --- |
| | | | Self-care | Educational | Patient support and motivation | | Availability | Details | |
| 9. | Tsai et al. [19] | iCKD | i. Blood pressure ii.Weight iii.Exercise iv.Meal photo v. Medication compliance vi. Blood test: SMBG , blood urea nitrogen (mg/dL) eGFR (ml/min/1.73 m2) hemoglobin (g/dL) albumin (g/dL) uric acid (mg/dL) cholesterol (mg/dL) triglyceride (mg/dL) urine protein/creatinine ratio (mg/mg) HbA1c (%) | Educational video i.Kidney anatomy & physiological functions, ii.Symptoms of CKD, and iii. Diagnostic indicators such as blood pressure and laboratory values (e.g., eGFR, hemoglobin) | Education modules, self-care feedback, reminder alerts | None reported | Yes | Moderate—tracking dashboard accessible by clinic staff to review motivation and biometric data. | A1c, Motivation scores |
| 10. | Berlot et al. [20] | Diabetes Xcel mobile app | None | Animated videos + modules i. "What are the different types of diabetes and who is at risK?" ii. "Hyperglycemia and diabetes-related problems" iii. "How can I manage my diabetes with the ABCs?" iv. "How can I plan what to eat?" v. "Diabetes and exercise" vi. "How can I monitor my diabetes control? vii. "Diabetes medications" viii. "Tests your health care provider will do" ix. "Resources" | Supportive push messages, video-based coaching, reminders | None reported | Yes | Minimal—educational prompts without two-way provider communication. | Adherence, knowledge gain |
| 11. | Gunawardena et al. [21] | Smart Glucose Manager | i. Diet reminder ii. Exercise at user-defined times iii. Medication reminders iv. SMBG | None | Automated reminders, bolus insulin calculator, self-care tracker | None reported | Yes | High integration—telemedicine-based model with real-time counseling chat and provider alerts. | Glucose control, Patient satisfaction |

Educational content was incorporated in seven studies [11,15–20], delivered through interactive modules, videos, or structured text, supporting patient understanding of disease processes and lifestyle management. In contrast, performance incentives such as gamification or reward systems were reported in only two studies which one was conducted in United States [15] and the other in China [17].

## Supportive devices used

In addition to the use of basic mobile applications, a limited number of studies incorporated other supportive devices to enhance the functionality and user experience of self-management interventions. Out of the eleven studies reviewed, four reported the integration of additional tools. One study utilized a heart rate wristband to monitor physical activity parameters such as steps taken, calories burned, and sleep patterns [13]. Two studies employed the LINE app group function to deliver dietary and exercise information and to facilitate teleconsultations [13,16]. Another two studies incorporated Bluetooth-enabled glucometers, allowing users to synchronize blood glucose data directly with the mobile application for continuous monitoring [14,21]. Study 6 [16] integrated communication features, including chat messaging and video calls, to provide real-time interaction and support. The remaining studies did not report the use of external supportive devices and relied solely on the mobile application's built-in features.

## Integration with care team and outcome measures

Among all the 11 studies, only a minority demonstrated active integration between the mobile apps and healthcare providers. Some apps (e.g., DialBetesPlus and SMARTDiabetes) synchronized data with electronic medical records or allowed care teams to monitor patient progress [14,17]. Others (e.g., e-KidneyCare) used built-in alerts to notify providers of critical lab results or symptoms [12]. However, most apps were standalone tools, lacking bidirectional communication or real-time clinician input. The included studies reported a range of outcome measures encompassing adherence metrics (such as medication compliance and goal tracking), biometric improvements (including blood pressure, HbA1c, and weight), and patient-reported outcomes related to usability and satisfaction [11,21]

## Discussion

### Principal findings

This scoping review mapped the functional features of mobile applications designed to support self-management among individuals with DKD, T2DM, and CKD. From 3,521 records, eleven studies were included, reflecting increasing interest in digital self-management tools across diabetes and kidney-related conditions. Across these apps, four core self-management domains were identified: patient support and motivation, self-care monitoring, educational components, and performance incentives. Patient support and monitoring features predominated, while incentives and clinical integration remained limited.

### Self-management feature domains

Patient support and motivational features were consistently incorporated across all reviewed apps. Common elements included reminders, personalized feedback, messaging, and peer-support components, reflecting a shift toward interactive and patient-centred app design. Such features are intended to enhance treatment adherence, reinforce healthy behaviours, and sustain engagement, consistent with prior evidence linking digital support mechanisms with improved self-management behaviours in diabetes and kidney care [1,22].

Self-care monitoring was another highly prevalent domain, reported in most studies. Apps commonly enabled tracking of blood glucose, blood pressure, body weight, and physical activity, either manually or through wearable device integration such as Bluetooth-enabled glucometers. These monitoring functions align with established clinical practices for

managing diabetes-related complications and cardiovascular risk [2]. However, monitoring was often implemented as passive data collection, with limited linkage to individualized feedback or clinical decision support.

Educational components were present in approximately two-thirds of the apps, delivered through videos, structured modules, or quizzes. While these components aimed to improve disease knowledge, medication adherence, and lifestyle practices, their level of personalization varied substantially. Many apps relied on static educational content, highlighting a gap in adaptive learning approaches that respond to users' changing needs and disease progression [23,24].

Performance incentives, including gamification and reward mechanisms, were the least commonly implemented features, reported in only two studies. Despite growing evidence that incentives can support motivation and sustained engagement, their limited use suggests that this design element remains underexplored in CKD and DKD-focused applications [3,25].

### Supportive devices and integration with care teams

Only four apps incorporated wearable or supportive devices, and real-time, two-way communication with healthcare providers was described in just two studies. Although some apps enabled data sharing or automated alerts, most lacked interactive clinical integration. This represents a missed opportunity, particularly for chronic conditions requiring ongoing monitoring, medication adjustment, and multidisciplinary care. Integrated digital systems that facilitate structured communication between patients and healthcare teams may reduce care fragmentation and support continuity of care, as suggested by recent digital health evaluations [26,27].

### Regional and contextual variation

Variation in mobile app features appeared to reflect both the target population and geographic context. For example, apps developed for diabetes populations generally emphasized glycaemic monitoring and lifestyle-related education, whereas CKD-focused apps placed greater emphasis on symptom tracking, renal-specific monitoring, and general wellness. Apps targeting DKD combined features addressing both glycaemic and renal management, reflecting the more complex self-management needs of this group. Regional differences were also observed, with some Asian-developed apps more likely to incorporate wearable device integration and messaging features. Together, these findings suggest that both clinical priorities and local technological infrastructure influence how mobile app functionalities are designed and prioritized. [11,13–21]

### Outcome reporting versus effectiveness

Several studies reported outcomes related to adherence, glycaemic control, usability, and user satisfaction. However, this scoping review focused on mapping functional features rather than appraising effectiveness or synthesizing intervention effects. Heterogeneity in study designs, outcome measures, and follow-up durations limited meaningful comparison of effectiveness across studies. This descriptive approach is consistent with scoping review methodology and with prior reviews examining digital health app content and functionality [28].

### Data security and ethical considerations

Mobile health applications handle sensitive personal and clinical data, yet reporting on data security and ethical safeguards was limited across the included studies. Safe and trustworthy use requires clear attention to informed consent, secure data storage and transmission, appropriate access control, and transparency in how user data are collected and used. Ethical considerations also include ensuring equitable access and usability for individuals with lower digital literacy. These practices are aligned with international digital health governance and mobile security standards, which emphasize privacy protection and user trust [29–31]. Improving the reporting and implementation of these safeguards is important for the wider adoption of mobile apps in chronic disease self-management.

## Limitations

This review has several limitations. First, no formal quality appraisal was conducted, in line with Joanna Briggs Institute scoping review guidance, meaning that the methodological quality of the included studies may vary. Second, the review was limited to English-language publications indexed in selected databases, which may have resulted in the exclusion of relevant studies reported in other languages or in grey literature, such as app store descriptions, technical reports, or industry publications. Third, information on real-world app engagement, including download numbers, active users, and user retention, was not consistently reported in the included studies and therefore could not be analysed. In addition, details on app cost models (e.g., free, freemium, or subscription-based) were rarely reported, limiting assessment of affordability, equity, and practical use. Finally, this review focused on describing app features rather than evaluating clinical effectiveness. These limitations should be considered when interpreting the findings.

## Implications for app development and clinical practice

The findings of this review highlight several implications for future mobile health development and research. App developers should consider not only self-management features but also transparency regarding pricing, cost models, and long-term affordability to support equitable access. Greater reporting of real-world usage data, such as user engagement and retention, would also strengthen understanding of app sustainability and practical value. From a clinical perspective, mobile apps may be most useful when integrated into structured care pathways with clear roles for monitoring, feedback, and follow-up. Future research should move beyond feature mapping to include evaluation of implementation, user engagement, cost transparency, and equity outcomes, particularly for long-term conditions such as DKD and CKD.

## Conclusion

This scoping review mapped the core self-management features of mobile applications designed for DKD, DM and CKD populations. While patient support tools, self-monitoring capabilities, and educational components were widely implemented, few apps integrated motivational incentives or direct communication with healthcare providers. These findings highlight the importance of incorporating bi-directional data sharing between users and healthcare providers to support real-time clinical oversight and personalized feedback. Additionally, the inclusion of behavioral reinforcement strategies such as adaptive goal setting and incentive-based features can enhance user motivation and long-term engagement. Insights from this review may serve as a valuable resource for digital health developers, nephrology and diabetes care teams, and health policymakers aiming to design more patient-centered, functionally integrated mobile health interventions.

## Supporting information

**S1 Dataset. Extracted study-level data used to generate all tables and figures included in this scoping review.**
(XLSX)

**S2 Table. PRISMA-ScR checklist.**
(DOCX)

**S3 Table. Full electronic search strategy.**
(DOCX)

## Author contributions

**Data curation:** Hasmawati Yahya, Nani Draman, Najib Majdi Yaacob.

**Methodology:** Hasmawati Yahya, Azidah Adbul Kadir, Najib Majdi Yaacob.

 

**Project administration:** Nani Draman.

**Resources:** Nani DRAMAN.

**Writing – original draft:** Hasmawati Yahya.

**Writing – review & editing:** Nani Draman, Azidah Adbul Kadir, Najib Majdi Yaacob.

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
