## [Decision Letter · Decision Letter 0]

7 Nov 2025

Dear Dr. DRAMAN,

Thank you for submitting your manuscript to PLOS ONE. After careful consideration, we feel that it has merit but does not fully meet PLOS ONE’s publication criteria as it currently stands. Therefore, we invite you to submit a revised version of the manuscript that addresses the points raised during the review process.

**ACADEMIC EDITOR:**The submission reflects scientific relevance as adjudged by the reviewers. However, some fundamental issues limit its quality for publication in the current form. For instance, the absence of adequate discussion of the results with appropriate references has watered down the reproducibility of the methodologies, results and the overall chance for publication of the current submission. Additionally, the authors need to justify the significance of the study, relate it to the literature and identify the gap in the existing knowledge that this study aims to address. Again, what are the limitations of this study, and how can the authors recommend future research on the study? Moreover, some other major concerns have been raised by the reviewers affecting pivotal sections of the study. Kindly pay close attention to these and address them critically before resubmission.. 

We look forward to receiving your revised manuscript.

Kind regards,

Yusuf Oloruntoyin Ayipo, Ph.D

Academic Editor

PLOS ONE

Journal Requirements:

“This work was supported by the Universiti Sains Malaysia, Research University

Team (RUTeam) Grant Scheme with Project No: 1001/PPSP/8580083, Project Code:

TE0034 (Reference No: 2022/0501]

The funding provided covered the author’s role as a research assistant and facilitated access to the Universiti Sains Malaysia library’s electronic databases and resources”

“This work was supported by the Universiti Sains Malaysia, Research University

Team (RUTeam) Grant Scheme with Project No: 1001/PPSP/8580083, Project Code:

TE0034 (Reference No: 2022/0501]”

“This work was supported by the Universiti Sains Malaysia, Research University

Team (RUTeam) Grant Scheme with Project No: 1001/PPSP/8580083, Project Code:

TE0034 (Reference No: 2022/0501]

The funding provided covered the author’s role as a research assistant and facilitated access to the Universiti Sains Malaysia library’s electronic databases and resources”

Additional Editor Comments (if provided):

The submission reflects scientific relevance as adjudged by the reviewers. However, some fundamental issues limit its quality for publication in the current form. For instance, the absence of adequate discussion of the results with appropriate references has watered down the reproducibility of the methodologies, results and the overall chance for publication of the current submission. Additionally, the authors need to justify the significance of the study, relate it to the literature and identify the gap in the existing knowledge that this study aims to address. Again, what are the limitations of this study, and how can the authors recommend future research on the study? Moreover, some other major concerns have been raised by the reviewers affecting pivotal sections of the study. Kindly pay close attention to these and address them critically before resubmission.

Reviewers' comments:

Reviewer's Responses to Questions

**Comments to the Author**

1. Is the manuscript technically sound, and do the data support the conclusions?

Reviewer #1: Yes

Reviewer #2: Yes

Reviewer #3: Yes

Reviewer #4: Yes

Reviewer #5: Yes

2. Has the statistical analysis been performed appropriately and rigorously?

Reviewer #1: Yes

Reviewer #2: Yes

Reviewer #3: Yes

Reviewer #4: Yes

Reviewer #5: Yes

3. Have the authors made all data underlying the findings in their manuscript fully available?

Reviewer #1: Yes

Reviewer #2: Yes

Reviewer #3: Yes

Reviewer #4: Yes

Reviewer #5: Yes

4. Is the manuscript presented in an intelligible fashion and written in standard English?

Reviewer #1: Yes

Reviewer #2: Yes

Reviewer #3: Yes

Reviewer #4: Yes

Reviewer #5: Yes

Reviewer #1: Dear authors: After reviewing your manuscript, I found that it follows the template and meets the PLOS ONE style requirements, including file naming. The PLOS ONE style template can be found at:

The presentation of research data should be well-synthesized, with the aim of summarizing the number of pages and displaying it with attractive tables and graphs, following the PLOS ONE template.

Therefore, authors should further enhance their literacy by reading high-impact journals, as they reflect the quality of their research and publications.

Reviewer #2: Dear authors,

Thank you for your well-structured and timely review on mobile applications for diabetic kidney disease self-management. With diabetes and its renal complication being a major clinical and public health burden, your work will play a pivotal role in both management and prevention of diabetes-related kidney diseases.

Following my review, I have identifed a few areas that require your attention in other to Improve clarity and enhance impact.

1. Grammatical error and Language Inconsistency

Please ensure tense consistent (kindly review every section of this work with this point in mind), correct few grammatical errors, and vet included sentences for redundancies.

i. "However, because DKD is closely linked to both diabetes and CKD, mobile apps built for either condition often include overlapping features...”

Concern: Overly repetitive of the idea in subsequent sentences.

ii. “Looking across all three conditions helps provide a complete picture...”

Concern: "Looking across..." Please use a scientific language.

iii. “Studies was included if they describe...”

Concern: Review the your choice of verbal in relation to the plural subject, "Studies".

iv. “Two reviewers have independently screened...”

Copncern: You inclusion of "have" was not necessary.

v. “Table 3 shows four recurring core functional domains of self-management features were identified…”

Concern: Awkward phrasing.

2. Lengthy Discussion section.

I consider the discussion paragraphs unnecessarily lengthy and wordy. Kindly condense it.

3. The claim in this statement, “Our study is among the first to comprehensively explore mobile health apps spanning DKD, T2DM, and CKD.” is considered an overstatement because of your use of he word "first". Please, substitute with other words, e.g. "few"

Thank you once again for the privilage to review your submitted manuscript.

Reviewer #3: Thank you for the opportunity to review this manuscript which is of high public and clinical relevance.

In your work titled, which is a scoping review on the features of mobile apps supporting self-management among individuals with diabetic kidney disease, diabetes, and chronic kidney disease, you demonstrated a good command of scoping review methods and gave a vital descriptive synthesis of the functionalities of existing app. However, there are major issues in the Methodology and Discussion that will need further attention.

Methodology.

Issue 1. Absence of pilot testing which has affected Reproducibility.

Refer to this statement, "Data charting was conducted using a standardised extraction form developed by the review team, which included predefined fields based on the review objectives and PCC framework."

Recommendation: Please include a well coordinated pilot testing or validation to enhance reproducibility.

Issue 2. Data synthesis.

Recommendation: Please include a description of how features were categorized into the four functional domains.

DIscussion.

Issue 1: Structural concern.

I consider this section's paragraphs to be too long.

Recommendation: Please split this by theme; e.g. Self-management features, Integration with care teams, Regional variations, Limitations and implications. This will potentially increase depth of discussion but you do need to avoid wordiness.

Issue 2: Shallow limitation and Implications discussion.

Recommendation: Kindly include a subsection that will summarize the implications of your study for app developers and clinical practice. Similarly, expand on limitations of database coverage.

Issue 3: Data security and ethical implications.

Recommendation: Please expand on how data security and ethical concerns can be safely protected.

Issue 4: Minor tautology.

“Interactive, behaviorally informed, and patient-centred app design”

Recommendation: Kindly omit one of “behaviorally informed” and “patient-centred” as they have semantic overlap.

Issue 5: Potential Inconsistency.

Earlier, you stated under the subheading, Integration with Care team and Outcome Measure: "The included studies reported a range of outcome measures encompassing adherence metrics (such as medication compliance and goal tracking), biometric improvements (including blood pressure, HbA1c, and weight), and patient-reported outcomes related to usability and satisfaction [11,21]". However, this statement in the Discussion section, "Although outcomes such as adherence, glycemic control, or usability were reported in several studies, this review did not evaluate their effectiveness" can be seen as a contradiction. This is because the indices alluded to are key effectiveness indicators.

Grammatical Errors.

I encourage you to go through the whole manuscript and carefully look out for typographical errors, missed areas of punctuation, wordiness, awkward phrasing and overstatement with the aim of correcting them.

Reviewer #4: The work was done well, just a few corrections, In table 4, kindly stay consistent with how your sentences are written; that is, are you using lowercases all through or title cases? Therefore ensure consistency. spell check: there was a point where the word "through" is written as "thru". Lastly, what are the other methods/frameworks for conducting scoping reviews, and what influenced your decision for choosing the framework that was used in this review?.

Reviewer #5: The submitted scoping review is well-written and scientifically sound. This review successfully identifies and synthesizes the key features of mobile applications for self-management in DKD, DM, and CKD. The methodology appears robust, and the analysis provides a valuable contribution to the field of digital health.

It is clear and focuses well on identifying features of apps for kidney disease and diabetes; however, including data on downloads or active users would show which apps are truly engaging and have a proven track record. Also, clarifying the cost models of the reviewed apps (e.g., free, freemium, one-time purchase, subscription-based) is essential for assessing equity and practical utility. This information could be added as a new section in the results, or discussed as a key point for future research in the conclusion. Acknowledging this as a limitation also shows the authors have thought carefully about the bigger picture.

A minor editorial point was noted regarding citation style. It is recommended to use an en dash (–) for ranges of three or more consecutive numbers (Line # 172, #207, #250, #255).

**Do you want your identity to be public for this peer review?** For information about this choice, including consent withdrawal, please see our Privacy Policy

Reviewer #1: No

Reviewer #2: **Yes:** Kirean Kelechi Eze

Reviewer #3: **Yes:** Jessica Tochukwu Nzeadibe

Reviewer #4: No

Reviewer #5: No

---

## [Author Response · Author response to Decision Letter 1]

13 Feb 2026

We have carefully addressed all comments and suggestions provided by the Academic Editor and reviewers. A detailed, point-by-point response to each comment has been provided in the uploaded “Response to Reviewers” document. All revisions have been incorporated into the manuscript, and changes are indicated in the tracked-changes version.

---

## [Decision Letter · Decision Letter 1]

6 Mar 2026

Features of mobile apps for diabetic kidney disease self-management: a scoping review

PONE-D-25-44415R1

Dear Dr. DRAMAN,

We’re pleased to inform you that your manuscript has been judged scientifically suitable for publication and will be formally accepted for publication once it meets all outstanding technical requirements.

Kind regards,

Yusuf Oloruntoyin Ayipo, Ph.D

Academic Editor

PLOS One

Additional Editor Comments (optional):

The submission meets the level of scientific rigour required for publication in this title, and all the concerns raised by the respective reviewers have been addressed satisfactorily. I hereby recommend the manuscript for publication in the current version.

Reviewers' comments:

Reviewer's Responses to Questions

**Comments to the Author**

Reviewer #4: All comments have been addressed

Reviewer #5: All comments have been addressed

2. Is the manuscript technically sound, and do the data support the conclusions?

Reviewer #4: Yes

Reviewer #5: Yes

3. Has the statistical analysis been performed appropriately and rigorously?

Reviewer #4: Yes

Reviewer #5: No

4. Have the authors made all data underlying the findings in their manuscript fully available?

Reviewer #4: Yes

Reviewer #5: Yes

5. Is the manuscript presented in an intelligible fashion and written in standard English?

Reviewer #4: Yes

Reviewer #5: Yes

Reviewer #4: All corrections were adequate and addressed. This shows good academic resilience. Well done, authors.

Reviewer #5: (No Response)

**Do you want your identity to be public for this peer review?** For information about this choice, including consent withdrawal, please see our Privacy Policy

Reviewer #4: **Yes:** OLAMIDE AWOLAJA

Reviewer #5: No

---

## [Editor Report · Acceptance letter]

PONE-D-25-44415R1

PLOS One

Dear Dr. DRAMAN,

I'm pleased to inform you that your manuscript has been deemed suitable for publication in PLOS One. Congratulations! Your manuscript is now being handed over to our production team.

Kind regards,

on behalf of

Dr. Yusuf Oloruntoyin Ayipo

Academic Editor

PLOS One